# The Link between Hypermetabolism and Hypernatremia in Severely Burned Patients

**DOI:** 10.3390/nu12030774

**Published:** 2020-03-15

**Authors:** Christopher Rugg, Mathias Ströhle, Stefan Schmid, Janett Kreutziger

**Affiliations:** Department of General and Surgical Critical Care Medicine, Innsbruck Medical University Hospital, Anichstr. 35, 6020 Innsbruck, Austria; mathias.stroehle@tirol-kliniken.at (M.S.); stefan.schmid@tirol-kliniken.at (S.S.); janett.kreutziger@tirol-kliniken.at (J.K.)

**Keywords:** hypernatremia, hypermetabolism, burn, electrolyte-free water clearance

## Abstract

Hypernatremia is common in critical care, especially in severely burned patients. Its occurrence has been linked to increased mortality. Causes of hypernatremia involve a net gain of sodium or a loss of free water. Renal loss of electrolyte-free water due to urea-induced osmotic diuresis has been described as causative in up to 10% of hypernatremic critical ill patients. In this context, excessive urea production due to protein catabolism acts as major contributor. In severe burn injury, muscle wasting occurs as result of hypermetabolism triggered by ongoing systemic inflammation. In this retrospective study, severely burned patients were analysed for the occurrence of hypernatremia and subsequent signs of hypermetabolism. The urea: creatinine ratio—as a surrogate for hypermetabolism—sufficiently discriminated between two groups. Four of nine hypernatremic burn patients (44%) had a highly elevated urea: creatinine ratio, which was clearly associated with an increased urea production and catabolic index. This hypermetabolism was linked to hypernatremia via an elevated urea- and reduced electrolyte-fraction in renal osmole excretion, which resulted in an increased renal loss of electrolyte-free water. In hypermetabolic severely burned patients, the electrolyte-free water clearance is a major contributor to hypernatremia. A positive correlation to serum sodium concentration was shown.

## 1. Introduction

Hypernatremia is not only common in critical care patients [1] but has also been shown to be an independent risk factor for increased mortality [2,3]. In severely burned patients the incidence of hypernatremia is higher than in other intensive care populations (~35% vs. ~10%) [2,4]. Here also, together with an increased sodium variability, there has been a link to increased mortality up to 30–60% [5,6,7].

Causes of hypernatremia in critically ill patients can be manifold, but its development is either a result of a gain of sodium or a loss of free water [8]. Electrolyte-free water loss through osmotic diuresis due to urea has been described to be causative in 10% of hypernatremic critical ill patients [9]. To our knowledge, besides this retrospective analysis, further literature on this topic is scarce. All state a similar pathophysiology: an excessive generation of urea, usually due to catabolism deriving from various causes (e.g., diabetes type I, corticosteroids, hypermetabolism, systemic inflammation) leads to a profuse urine urea output and therefore an increased fraction of urea and decreased fraction of electrolytes in osmole excretion. This raised loss of electrolyte-free water is often overlooked and leads to hypernatremia [9,10,11,12,13].

In burn injury, the systemic inflammatory, as well as systemic stress, response leads to hypermetabolism, catabolism and muscle wasting [14]. This hypermetabolic response is concomitant with increased energy expenditure and energy substrate release from muscle protein and fat stores [15]. Protein is catabolized, leading to muscle wasting and excessive urea generation [16,17,18].

The aim of this retrospective analysis is to determine the role of hypermetabolism and concomitant catabolism as a cause of hypernatremia in burn patients treated in our department. Special attention will be paid to the assessment of the patient’s metabolic state. Our hypothesis is that burn-injury induced hypermetabolism is linked to hypernatremia via urea induced osmotic diuresis and an accompanying increased renal loss of electrolyte-free water. Furthermore, we suspect an elevated urea: creatinine ratio to predict such hypermetabolism, accompanied by an elevated urea generation rate, energy expenditure and catabolic index.

## 2. Materials and Methods 

This retrospective study was approved by the Ethics Committee of the Medical University of Innsbruck (EK Nr: 1269/2019) and the Institutional Review Board.

The department for general and surgical critical care medicine of the Innsbruck Medical University Hospital consists of two intensive care units (ICU), containing 23 level 3 intensive care beds treating multiple trauma and burn injury patients and patients following scheduled and emergency cardiac, vascular, thoracic and abdominal surgery as well as transplantations. Of approximately 650–700 patients per year, 5–10 patients suffer from severe burn injuries. 

For this analysis the local ICU database was queried for the diagnosis “combustio“ and/or “burn” for the time period January 1, 2017 to October 31, 2019. Inclusion criteria were non-pregnant patients over 18 years of age suffering from severe burn injuries, defined as the occurrence of at least one of the following criteria: >20% total body surface area (TBSA) affected, >5% full thickness burn or concurrent inhalational trauma. Additional criteria were an ICU length of stay of over seven days and at least one phase of hypernatremia (>145 mmol/l) under intact renal function (creatinine-clearance >30 mL/min; no need for renal replacement therapy). The query primarily resulted in *n* = 21 patients, from whom ten were excluded due to a length of stay under seven days (two died, two were repatriated early and six were transferred to a normal ward, doing well). Another two were excluded due to renal impairment and requirement of renal replacement therapy prior to any possible hypernatremia. From the remaining *n* = nine patients, two patients also required renal replacement therapy but not until after the included phase of hypernatremia. One patient required renal replacement therapy during serum sodium (NaS) decline. This phase (with creatinine clearances <30 mL/min) was excluded separately. Since renal function was intact during sodium incline—the main reporting period—the patient was kept for analysis.

Medical information was obtained by searching the local hospital information system as well as the ICU patient data management system (PDMS; GE, Centricity Critical Care 9.0 SP1). Parameters of interest were age, weight, SAPS-III admission score and ICU length of stay (LOS), NaS as assessed by direct potentiometry, plasma osmolality (Osmo_Pl_), fluid balance, urine output (OUT_Ur_), serum creatinine (Creas) and urea (Urea_s_), serum glucose concentration (Gluc_S_), 24-h urine lab (sodium- (Na_Ur_), potassium- (K_Ur_), creatinine- (Crea_Ur_), urea-concentration (Urea_Ur_), osmolality (Osmo_Ur_)), ventilator derived VCO_2_, calorie intake and composition (protein, nitrogen (IN_N_), carbohydrates, fat), signs of systemic inflammation (CRP, procalcitonin, leukocytes, body temperature) and any use of diuretics, glucocorticoids or insulin. Derived or calculated variables included sodium balance as difference between total sodium intake and urine sodium output, creatinine- (CrCl), free water- (FWC) and electrolyte-free water clearance (EFWC), urine urea excretion, rate of urea generation (UreaGR), catabolic index, serum urea to creatinine ratio (Urea:Crea, both as mg/dl), ratio of urine osmolality due to urea vs. non-urea osmoles (Osmo_Urea_:Osmo_Rest_), energy expenditure (EE_VCO2_) and nitrogen balance derived from protein intake and urine urea output. Urea generation rate was defined as total amount of renal urea excretion added to the amount of serum urea changes. These were estimated by multiplying changes in serum urea concentration with expected total body water (0.6 * body weight). Bistrian described a catabolic index in 1979 by utilizing renal nitrogen excretion as marker for ureagenesis and subtracting half of the total nitrogen intake as impact of dietary protein intake as well as an additional 3 g as approximation for other obligatory nitrogen losses [19]. Negative to slightly positive values indicate no to moderate stress, as urea production can be explained by dietary intake and obligatory losses. The higher the value, however, the higher metabolic stress as well as protein catabolism is assumed. A modified Bistrian catabolic index was computed by using urea generation rate rather than merely urine urea excretion. Energy expenditure was derived from VCO_2_ based on an assumed respiratory quotient of 0.86 as described by Stapel et al. in 2015 [20]. Free and modified electrolyte-free water clearances were calculated according to the formulas described by Nguyen and Kurtz in 2012 [21]. 

Formulas, corresponding units and references are shown in Table 1. 

Sodium content of IV-fluids and drugs—as per product information sheet—is stored in our PDMS; so are calories and their composition (protein, nitrogen, fat, carbohydrates) from enteral and parenteral feeding as well as non-nutritional calories from propofol and glucose solutions for drug administration. Total intake of these components is therefore generated and balanced automatically.

Regarding data analysis, as a first step data of included patients were examined for signs of excessive hypermetabolism, as we hypothesized that this was potentially able to affect hypernatremia due to urea induced osmotic diuresis. As recently published, an elevated serum urea:creatinine ratio above 75 (75 mg/dl:mg/dl = 141 mmol/l:mmol/l) was a marker of hypermetabolism [22]. By utilising this ratio during hypernatremic phases, a division into two groups was possible. The two groups were then primarily compared with regards to the above-mentioned parameters during incline in NaS (Table 2, Figure 1), as well as separately from admission up to day 25 (day of latest NaS peak; Figure 2). Data were tested for normal distribution by Shapiro-Francia test and presented as median and interquartile range (Q1–Q3) due to non-normal distribution. In this nonparametric setting the two independent groups consisting of multiple, intraindividual longitudinal data were analysed using a modified ANOVA-type statistic with Box approximation (R package: nparLD; Table 2, Figure 1 and Figure 2) [23]. A visual correlation analysis was performed by the repeated measures correlation method described by Bakdash et al. (R package: rmcorr; Figure 3) [24]. This technique is an atypical application of ANCOVA and is based on assumptions also required for general linear modelling, among them normal distribution of errors. For this reason, we did not present an exact correlation coefficient and p-value for the correlation analysis. Regarding demographics, Fisher’s exact test was performed to detect group differences in frequencies and Mann Whitney U-test for group differences of continuous data (Table 3). A *p*-value < 0.05 was considered significant.

## 3. Results

### 3.1. Demographics

Setting the urea: creatinine ratio threshold during hypernatremia to 75 yielded two groups with four patients in the *high* group (above 75) and five in the *low* group (below 75). Age, burn injury severity, SAPS III, ICU LOS and overall ICU outcome were comparable between the groups. While the duration of NaS incline as well as the absolute height of NaS peak were comparable, NaS peak was reached later (LOS day 20.0 vs. 11.0) in the *high* group. General demographics are presented in Table 3.

### 3.2. Hypermetabolism

During NaS incline, urea: creatinine ratio was 81.3 in the *high* and 48.5 in the *low* group and was clearly associated with other signs of hypermetabolism (Table 2, Figure 1). The catabolic index as well as urea generation rate were also significantly elevated in the *high* group: 9.3 vs. 5.9 and 54.9 g/d vs. 35.8 g/d respectively. Insulin requirements, as well as other signs of ongoing systemic inflammation, e.g., CRP, leukocytes and body temperature, all tended to be more elevated in the *high* group but did not reach statistical significance. There was no use of glucocorticoids, as possible cause of catabolism, in both groups (Table 2).

Time dependent analysis from admission up to ICU-day 25 revealed a similar course for calorie and weighted protein intake as well as nitrogen balance in both groups (Figure 2). Although an elevated trend in VCO2-derived energy expenditure can be seen graphically, statistical analysis was restricted due to a majority of extubated patients in later phases of the *low* group.

### 3.3. Electrolyte-Free Water Clearance

The increased urea generation rate in the *high* group was also accompanied by a raised fraction of urea in urine osmolality when compared to other osmoles – Osmo_Urea_: Osmo_Rest_ 1.1 vs. 0.6 during NaS incline (Table 2, Figure 1). Reflecting this elevated ratio, urine sodium- and potassium-concentrations—as major determinants of urine Osmo_Rest_—were also significantly reduced in the *high* group (62.0 mmol/l and 35.0 mmol/l vs. 100.0 mmol/l and 64.5 mmol/l respectively). At the same time patients in the *high* group were more polyuric than in the *low* (3840 mL/d vs. 1995 mL/d). Consequently, the loss of electrolyte-free water in terms of the EFWC differed highly significantly between the groups (Figure 1). The *high* group lost a median of 1645 mL and the *low* group merely 27 mL of electrolyte-free water daily during NaS incline.

Furthermore, visual correlation analysis from beginning to end of NaS decline showed a positive linear correlation between EFWC and NaS in the *high* and no correlation in the *low* group for the observed hypernatremic timeframe (Figure 3). Being clearly negative in both groups (-1358 mL *high* vs. -2133 mL *low*), the free water clearance was misleading, as it indicated persistent water retention during incline in NaS. Creatinine clearance was high during NaS incline, showing sufficient renal function with no differences between the groups (98.0 mL/min vs. 119.5 mL/min).

### 3.4. Hypernatremia

In respect of other major determinants of hypernatremia, total sodium as well as fluid balances and use of diuretics were examined. From admission up to ICU day 25 (latest NaS peak) daily sodium as well as fluid balances did not differ between the groups (Figure 2). As expected, sodium balances were highly positive in both groups initially during the resuscitation phase and then flattened out to become nil, even to slightly negative. Fluid balances were clearly positive during the total timeframe and especially during NaS incline phases.

During incline in NaS, furosemide and canrenone were used in both groups without significant difference and in rather low doses (Table 2).

## 4. Discussion

Four of nine hypernatremic burn patients (44%) had a highly elevated urea: creatinine ratio, which was clearly associated with an increased urea production and catabolic index. This hypermetabolism was linked to hypernatremia via an elevated urea- and reduced electrolyte-fraction in renal osmole excretion. The resulting increased renal loss of electrolyte-free water is often underestimated, as classical free water clearances and fluid balances are misleading in these situations. In hypermetabolic severely burned patients, the electrolyte-free water clearance is a major contributor to hypernatremia, especially when onset is later. A positive correlation to serum sodium concentration was shown.

### 4.1. Demographics

By searching the ICU database in the given timeframe, 21 burn patients were extracted, whereof nine were included. Two of the included (one in each group) required renal replacement therapy at some point after the included hypernatremic phase. One patient (high group) required renal replacement therapy during decline of NaS. Despite this phase being excluded separately from further analysis, the patient was kept included due to the fact that renal function was not impaired during NaS incline, the main reporting period. Creatinine clearances of both groups during NaS incline had a median and interquartile range of 109 (76.5–160) ml/min and a minimum of 47 mL/min. Creatinine clearances during included NaS decline phases were always greater than 30 mL/min. The incidence of hypernatremia in burn patients treated at our department was 9 of 21 (43%) and therefore higher than described elsewhere [4,5,6]. Increased age, burn extent, mechanical ventilation and ventilation duration have been shown to be independent predictive factors for hypernatremia [5]. Besides mean age being higher in our cohort than in the others mentioned, burn extent was rather lower. Interestingly a burn-cohort from our department, from over thirty years ago, also showed lower incidences of hypernatremia despite suffering from more severe burn injuries [25]. A mixture of 5% glucose solution with 20% albumin and 80 mmol of sodium (as bicarbonate or chloride) was formerly used as resuscitative fluid following the modified Baxter formula. Meanwhile we relied on balanced crystalloids or colloids (gelatin) administered according to dynamic values as in pulse pressure or stroke volume variation as well as urine output. Therefore, we suspect that the main differences lie in initial fluid resuscitation and general fluid balance handling. Whether a more restrictive fluid handling at a cost of a higher incidence of hypernatremia is beneficial or not cannot be said at this point. 

### 4.2. Hypermetabolism

Hypermetabolism was primarily screened for using the serum urea: creatinine ratio during hypernatremia. An elevated ratio has recently been described to represent catabolism and muscle wasting in major trauma patients [22]. Patients with prolonged lengths of stay (≥10 days) showed a significant increase in the urea: creatinine ratio when compared to baseline as well as patients dismissed sooner. This elevated ratio was linked to catabolism by CT evaluation of lumbar muscle cross sectional area. The median increased ratio observed in that retrospective study was 141 [*mmol/l:mmol/l*] which equates to the threshold used in this study. We were able to link this elevated ratio to other signs of hypermetabolism and protein catabolism, such as increased urea genesis and catabolic index. 

Slight modification was made to Bistrian’s catabolic index by using the urea generation rate rather than mere renal urea excretion, as we felt the index to be more accurate in this way as changes in UreaS are also accounted for. The index was not manipulated much by this change as over 90% of the UreaGR was attributed to renal urea excretion in this study. Severe stress, as in a very high catabolic index, was indicated for the *high,* and moderate to severe for the *low* group.

Regarding energy expenditure, ventilator derived VCO_2_ was utilized as described in a proof of concept study from 2015 [20]. After rewriting the Weir formula and computing a respiratory quotient on the basis of the composition of most popular nutritional products (RQ = 0.86), the above-mentioned formula was developed. The assumption that what is fed is equal to what is absorbed and, finally, to what is burned in substrate oxidation is surely the biggest limitation to this formula [26]. Malabsorption and endogenous substrate liberation (e.g., gluconeogenesis, proteolysis, lipolysis) are completely neglected. Albeit simplifying complex physiology, accuracy was surprisingly good, as a bias of merely 7.7% was observed when compared to the gold standard (indirect calorimetry).

Dividing the population by signs of excessive hypermetabolism also led to a non-significant discrepancy in gender distribution suggesting males to be more prone to hypermetabolism than females. Given the higher proportion of muscle mass in total body weight in males, a greater impact of muscle wasting on urea production would not seem surprising. However, studies have shown that gender differences go further than that. In a prospective cohort study, female pediatric patients showed an attenuated inflammatory, hypermetabolic and catabolic response to a severe burn when compared to male patients [27,28]. This was concomitant with higher levels of endogenous anabolic hormones and lower levels of stress hormones and inflammatory markers.

An increased urea production can be caused by metabolism of exogenous or endogenous amino acids [29,30,31]. The differentiation between the two possible pathogeneses is not easy to make and was not the aim of this study. Exact protein requirements are quite unclear in the setting of burn injury and protein provision to the observed study patients was within normal ranges [15]. It is, however, worth mentioning that the *high* group also received more protein than the *low* group. At the same time both groups had comparable nitrogen balances (Figure 2). The concomitant trend also of higher signs of systemic inflammation, as in CRP and leukocyte counts in the *high* group, could give a hint of ongoing hypermetabolism and endogenous catabolism (Table 2). Insulin requirements—as indirect sign of feeding intolerance—also showed an elevated tendency in the *high* group but was not excessively high in either group. Lastly, it remains unclear if the increased protein intake is a cause of elevated urea generation or is merely a therapeutic answer to suspected hypermetabolism.

### 4.3. Electrolyte-Free Water Clearance

Urea induced osmotic diuresis has been described to be causative in 10% of hypernatremic patients in a medical ICU environment [9]. In our cohort of severely burned patients we were able to link as much as 44% to this phenomenon. Additionally, there have only been a few case reports on this topic [9,10,11,12,13]. The corresponding pathophysiology is easily understandable but even more easily overseen. Independent of the underlying cause, an excessive urea production leads to an accompanied increase in urea in the urine. Depending on total amount of urea and the kidneys’ ability to concentrate, the osmole excretion becomes dominated by urea at the expense of electrolytes. The ratio of urea-induced urine osmolality to other osmoles rises, leading to a urine that is in fact hyperosmolar, but has low sodium and potassium concentrations. Regarding these NaS determining osmoles. the urine is actually dilute. Polyuria combined with low electrolyte content is what defines the osmotic diuresis and predicts electrolyte-free water loss and rising serum sodium at the same time. Commonly used free water clearance is calculated by comparing urine and plasma osmolality and is therefore misleading in these situations. Being constantly negative, it indicated concentrated urine and water retention in this study. While having this relatively concentrated, even hyperosmolar, urine, a renal loss of free water is certainly not additional to the differential diagnoses regarding hypernatremia. However, as seen in the *high* group, the electrolyte-free water clearance can be the key, as it was clearly positive, indicating constant loss of electrolyte-free water during NaS incline. Despite the limitation of these formulas, strictly describing renal handling and not taking input and extra-renal loss of sodium and water into account the *high* group showed a significant correlation between EFWC and NaS during NaS incline and decline. Extra-renal and insensible water losses must have been covered well by general fluid balances, which were constantly positive, making the EFWC the major contributor to hypernatremia in the hypermetabolic group. The missing correlation in the *low* group fits in with most slightly positive EFWC, and only shows that hypernatremia was not due to renal water losses. 

### 4.4. Hypernatremia

Hypernatremic states are reported quite often after initial fluid resuscitation of burn injuries [2,4,5,6,7]. Increased loss of total body water through insensible losses, combined with an initial total sodium burden due to high resuscitation fluid administration, are usually described as the main pathophysiology and can also be seen in our study (Figure 2). In a study conducted in 1973, cases of hypernatremia in burn patients were described as the result of combined osmotic diuresis induced by glucosuria and elevated nitrogen excretion due to hyperalimentation with amino acids [32]. Besides this report, renal water losses are apparently seldomly thought of in severely burned patients but can also contribute to hypernatremia as described in this study. In general, reasons for hypernatremia are manifold and certainly not easily broken down to one direct cause, although the result is always a net gain of sodium or loss of free water [8]. 

While duration of NaS incline was similar in both groups, the *high* group reached its sodium peak significantly later, while the *low* group started NaS incline earlier. As also confirmed in Figure 2, initial fluid resuscitation of burn patients allows higher fluid balances in the early phase of ICU care. On the other hand, during a later onset of hypernatremia initial fluid losses through wound secretions may be expected to be ceased. Excessive water losses are therefore not expected, at least not extra-renally. An increased sodium input through ongoing therapies (e.g., antibiotics, drug solutions and iv-fluids) seems also possible [33]. Hence the use of canrenone in the *high* group. Sodium balances, as difference between total sodium intake and renal sodium excretion, were also clearly positive initially in both groups (Figure 2) and did not differ between the groups regarding total time course from admission up to day 25. Explicitly during NaS incline they were merely slightly positive in both groups. With respect to the fact that extra-renal sodium losses (i.e., stool and wound excretions, etc..) are not taken into account one can assume sodium balances to actually be at least even during NaS incline. The use of furosemide was scarce and not different between the groups, so that its contribution to hypernatremia can be neglected. 

We conclude that renal water losses—despite hyperosmolar, relatively concentrated urine—are the major contributor to hypernatremia in the hypermetabolic group, whereas extra-renal water losses in combination with an elevated initial sodium burden are most probably causative for hypernatremia in the low group. While therapy is theoretically simple in both groups, aggravation of fluid balances to even more positive values is not always easy for an intensivist.

### 4.5. Limitations

The main limitation of this study is the small sample size which was caused by limited and incomplete data acquisition of our PDMS. Reasonable estimation of incidences is therefore not possible. The general informative value of a single centre retrospective study is also limited. Nevertheless, the results differed highly significantly between the groups, so that the pathophysiology of EFWC resulting in hypernatremia is very well presented. 

## 5. Conclusions

Electrolyte-free water clearance is a major contributor to later onset hypernatremia in hypermetabolic severely burned patients with intact renal function. In our analysis four of nine (44%) hypernatremic patients also had signs of profuse hypermetabolism. This state was sufficiently discriminated by an elevated urea: creatinine ratio. Excessive urea generation led to an elevated ratio of urea over electrolytes in osmole excretion and consequentially to an increased electrolyte-free water clearance. These losses are often underestimated as classical free water clearances and fluid balances are misleading in these situations. For severely hypermetabolic burn patients the EFWC was positively correlated to serum sodium concentrations.

## Figures and Tables

**Figure 1 nutrients-12-00774-f001:**
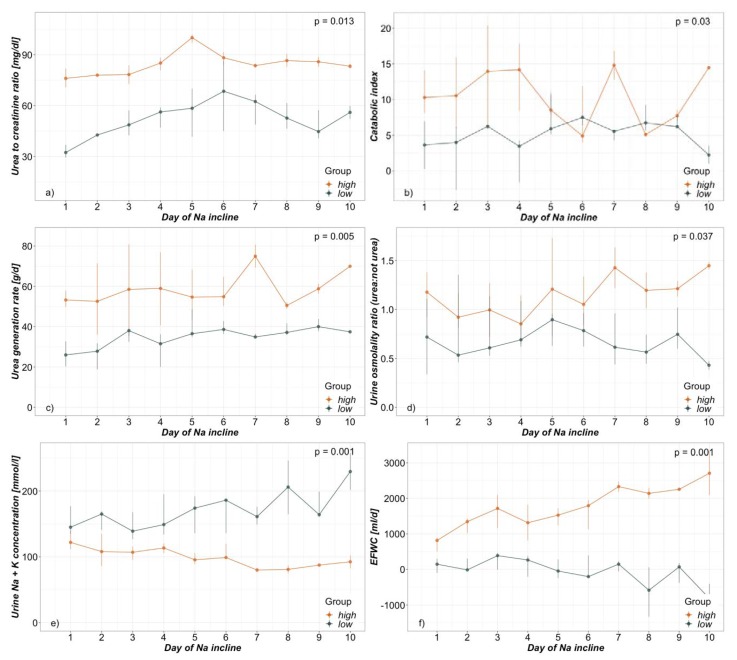
Day of NaS incline dependent group analysis presented as daily median and interquartile range of (**a**) Urea:Creatinine ratio (**b**) Catabolic index (**c**) Urea generation rate (**d**) Urine osmolality ratio (urea:not urea) (**e**) Sum of urine sodium and potassium concentrations and (**f**) electrolyte-free water clearance (EFWC).

**Figure 2 nutrients-12-00774-f002:**
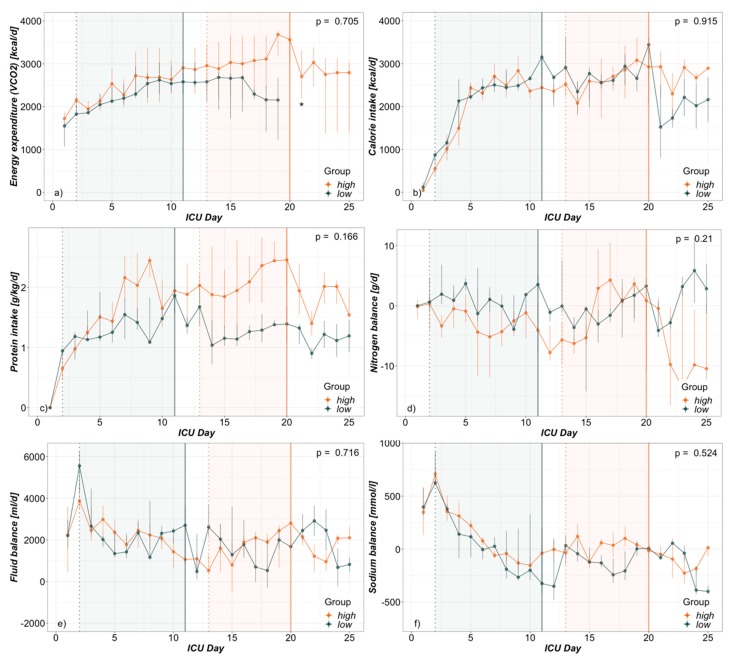
ICU-day dependent group analysis presented as daily median and interquartile range from admission to day 25 (day of latest NaS peak) of (**a**) Energy expenditure (**b**) Calorie intake (**c**) Protein intake (**d**) Nitrogen balance (**e**) Fluid balance and (**f**) Sodium balance. Vertical lines indicate median day of NaS peak (solid) and beginning of NaS incline (dotted). * incomplete values due to partially extubated patients.

**Figure 3 nutrients-12-00774-f003:**
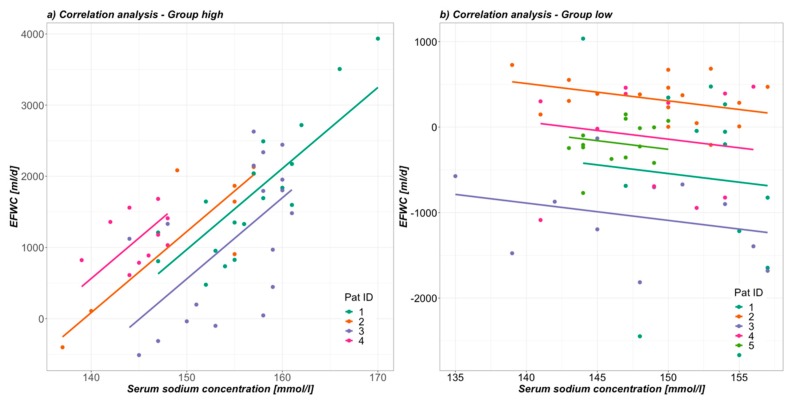
Correlation between EFWC and NaS during hypernatremic phase in (**a**) group *high* and (**b**) group *low*.

**Table 1 nutrients-12-00774-t001:** Formulas, corresponding units and references.

Formula	Units	Ref
$FWC=OUT_{UR}\times \left[1-\left(\frac{Osmo_{Ur}}{Osmo_{P}}\right)\right]$	[ml/d]	[21]
$EFWC=OUT_{UR}\times \left[1-\left(\frac{1.03\times (Na_{Ur}+K_{Ur})}{Na_{S}+23.8+0.016\times \left(Gluc_{S}-120\right)}\right)\right]$	[ml/d]	[21]
$UreaGR=OUT_{UR}\times Urea_{Ur}+\left(Urea_{S\left[today\right]}-Urea_{S\left[yesterd.\right]}\right)\times 0.6\times body\;weight$	[g/d]	
$Catabolic\;index=\frac{UreaGR}{2.14}-\left(0.5\times IN_{N}+3g\right)$	[g/d]	[19]
$EE_{VCO2}=VCO2\;\left[\frac{ml}{min}\right]\times 8.19$	[kcal/d]	[20]

**Table 2 nutrients-12-00774-t002:** Group characteristics during serum sodium (NaS) incline. (*n* indicating total days of NaS-incline).

Urea:creatinine	*High* (*n* = 31) Median (Q1–Q3)	*Low* (*n* = 44) Median (Q1–Q3)	*p*
Urea:creatinine	81.3 (78.4–89.0)	48.5 (40.0–66.9)	0.013
Catabolic index	9.3 (6.1–17.4)	5.9 (3.6–6.8)	0.030
Urea generation rate [g/d]	54.9 (43.7–72.6)	35.8 (31.7–39.1)	0.005
Insulin requirements [IU/d]	14.8 (0.0–45.6)	0.0 (0.0–24.6)	0.208
CRP [mg/dl]	25.8 (15.6–29.6)	19.9 (8.4–24.0)	0.220
Leucocytes [G/l]	10.1 (9.4–11.5)	7.3 (6.5–12.2)	0.376
Temperature [°C]	37.9 (37.2–38.3)	37.7 (37.0–38.2)	0.688
Urine Osmo_Urea_:Osmo_Rest_	1.1 (0.8–1.5)	0.6 (0.5–1.0)	0.037
Na^+^ concentration urine [mmol/l]	62.0 (51.0–79.5)	100.0 (62.8–147.5)	0.026
K^+^ concentration urine [mmol/l]	35.0 (28.0–49.5)	64.5 (51.8–82.8)	0.014
Electrolyte-free water clearance [ml/d]	1645 (824–2060)	27 (−239–316)	0.001
Free water clearance [ml/d]	−1358 (−2091–466)	−2133 (−2899–−1423)	0.061
Creatinine clearance [ml/min]	98.0 (71.0–148.0)	119.5 (88.0–164.3)	0.487
Urine output [ml/d]	3840 (3120–4410)	1995 (1380–2958)	0.024
Furosemide use [mg/d]	0.0 (0.0–83.3)	0.0 (0.0–92.4)	0.766
Canrenone use [mg/d]	0 (0–400)	0 (0–0)	0.479

**Table 3 nutrients-12-00774-t003:** Demographics.

Urea:creatinine	*High* (*n* = 4) Median (Q1–Q3)	*Low* (*n* = 5) Median (Q1–Q3)	*p*
Age [years]	55.5 (52.5–59.3)	42.0 (27.0–62.0)	0.539
Gender *male* *female*	3 1	2 3	0.524
Burn injury TBSA [%] *full thickness burn TBSA [%]* *inhalation injury [n]*	26.0 (24.0–28.8) 2.5 (0–5.3) 3	35.0 (20.0–40.0) 5.0 (4.0–9.0) 1	0.711 0.321 0.206
SAPS III admission score	52.5 (48.0–57.5)	52.0 (48.0 – 61.0)	0.901
ICU LOS [d]	45.0 (31.8–59.0)	31.0 (23.0 – 37.0)	0.219
ICU outcome *dead* *alive*	0 4	0 5	1.000
Duration of NaS-incline [d]	8.0 (5.8–10.0)	9.0 (8.0–10.0)	0.701
NaS peak [mmol/l]	159.0 (154.8–163.3)	157.0 (156.0–157.0)	0.443
LOS day of NaS peak	20.0 (18.5–22.0)	11.0 (10.0–15.0)	0.027

Data were analysed with RStudio version 1.2.5001 (RStudio, Inc., Boston, MA).

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
