# Peer review of "The Link between Hypermetabolism and Hypernatremia in Severely Burned Patients"

_nutrients, 2020, doi:10.3390/nu12030774_

Round 1
Reviewer 1 Report
Rugg et all described elegantly the link between hypermetabolism and hypernatremia in severely burned patients. This was implicated through elevated urea- and reduced electrolyte-fraction in renal osmole excretion. This study is important for the assessment of the patient's metabolic state and to improve their welfare.
Minor suggestion:
Authors could considered to correlate the study with this paper published recently: https://doi.org/10.1002/ncp.10390
Overall, it is a nicely written research article.
Author Response
Dear Reviewer,
We would like to thank you very much for your time and your positive review. We feel very complimented that are manuscript appeals to you.
We also must thank you for your hint to the mentioned paper. The concise review by Wise et al. couldn’t have fit better to our manuscript!
Changes are traceable by the Track-Changes function in the main text and marked in yellow in our answers below.
We decided to add this reference as [15] by adding or adapting the following sentences:
Line 44-45: “This hypermetabolic response is concomitant with increased energy expenditure and energy substrate release from muscle protein and fat stores [15].”
Line 277-278: “Exact protein requirements are quite unclear in the setting of burn injury and protein provision to the observed study patients was within normal ranges [15].“
Reviewer 2 Report
The manuscript by Rugg et al. focuses on the hypernatremia, in particular, in severely burned patients. Hypernatremia has been linked to increased mortality. In their retrospective study, the authors assessed severely burned patients for the occurrence of hypernatremia and subsequently signs of hypermetabolism. In addition, the urea:creatinine ratio – as a surrogate for hypermetabolism – was discriminated between two groups. Four of nine hypernatremic burn patients (44%) had a highly elevated urea:creatinine ratio, which was clearly associated with an increased urea production and catabolic index. This hypermetabolism was linked to hypernatremia via an elevated urea- and reduced electrolyte-fraction in renal osmole excretion, which resulted in an increased renal loss of electrolyte-free water. A positive correlation to serum sodium concentration was shown. The authors have supported their claims with clearly defined experiments and the manuscript is written well with apt background
Minor comments:
1. Lines 61-64: “For this analysis the local ICU database was queried for the diagnosis „combustio“ and/or 62 „burn” for the time period of January 1, 2017 to October 31, 2019. Inclusion criteria were non pregnant 63 patients over 18 years of age suffering from severe burn injuries (either > 20 % total body surface area 64 (TBSA) affected, over 5 % full thickness burn, or concurrent inhalational trauma) (…)”
What are the inclusion criteria for this study: either > 20 % total body surface area 64 (TBSA) affected, OR over 5 % full thickness burn, or concurrent inhalational trauma? Please explain it better.
2. Lines 25-256: “Nevertheless, it is noteworthy that the high group also received more protein than the low group by trend”.
What do the authors mean by this? Please explain it better.
3. Table 3. Group characteristics during NaS incline. (n indicating total days of NaS-incline).
Have the authors characterized the type of leucocytes?
Author Response
Dear Reviewer,
we would like to thank you for your time and your helpful comments!
We are pleased that our manuscript appeals to you.
We revised the main text of the manuscript step by step according to your comments. Changes are traceable by the Track-Changes function in the main text and marked in yellow in our answers below.
- Lines 61-64: “For this analysis the local ICU database was queried for the diagnosis „combustio“ and/or 62 „burn” for the time period of January 1, 2017 to October 31, 2019. Inclusion criteria were non pregnant 63 patients over 18 years of age suffering from severe burn injuries (either > 20 % total body surface area 64 (TBSA) affected, over 5 % full thickness burn, or concurrent inhalational trauma) (…)”
What are the inclusion criteria for this study: either > 20 % total body surface area 64 (TBSA) affected, OR over 5 % full thickness burn, or concurrent inhalational trauma? Please explain it better.
Dear Reviewer,
Thank you for pointing out this important potential misunderstanding. For a better understanding of the inclusion criteria we adapted the sentences as follows:
Line 63-67: “Inclusion criteria were non pregnant patients over 18 years of age suffering from severe burn injuries, defined as the occurrence of at least one of the following criteria: > 20 % total body surface area (TBSA) affected, > 5 % full thickness burn or concurrent inhalational trauma. Additional criteria were an ICU length of stay of over seven days and at least one phase of hypernatremia (> 145 mmol/l) under intact renal function (creatinine-clearance > 30 ml/min; no need for renal replacement therapy).
- Lines 25-256: “Nevertheless, it is noteworthy that the high group also received more protein than the low group by trend”.
What do the authors mean by this? Please explain it better.
Dear Reviewer,
Once again, we would like to thank you for your precise and critical reading. The sentence refers to the first sentence of that paragraph talking about an increased urea production being caused by endo- or exogenous amino acids. We wanted to point out that the high group - with a clearly elevated urea generation rate - also received more protein. Although nitrogen balances were comparable, it remains unclear “if the increased protein intake is a cause of elevated urea generation or is merely a therapeutic answer to suspected hypermetabolism” (line 297f).
We decided to rephrase the section as follows:
Line 277ff: “An increased urea production can be caused by metabolism of exogenous or endogenous amino acids [29–31]. The differentiation between the two possible pathogeneses is not easy to make and was not aim of this study. Exact protein requirements are quite unclear in the setting of burn injury and protein provision to the observed study patients was within normal ranges [15]. It is however worth mentioning that the high group also received more protein than the low group. At the same time both groups had comparable nitrogen balances (Figure 3).”
- Table 3. Group characteristics during NaS incline. (n indicating total days of NaS-incline).
Have the authors characterized the type of leucocytes?
Dear Reviewer,
We must agree that this could be highly informative.
Unfortunately, differential leukocyte count is not part of our routine, daily lab-workup, and only done on special request – usually when leukocyte count is very low or extremely high.
With leukocyte counts being normal to merely slightly elevated there were no such analyses.